# Feedforward Artificial Neural Network-Based Colorectal Cancer Detection Using Hyperspectral Imaging: A Step towards Automatic Optical Biopsy

**DOI:** 10.3390/cancers13050967

**Published:** 2021-02-25

**Authors:** Boris Jansen-Winkeln, Manuel Barberio, Claire Chalopin, Katrin Schierle, Michele Diana, Hannes Köhler, Ines Gockel, Marianne Maktabi

**Affiliations:** 1Department of Visceral, Transplant, Thoracic and Vascular Surgery, University Hospital of Leipzig, 04103 Leipzig, Germany; manuel.barberio@ihu-strasbourg.eu (M.B.); ines.gockel@medizin.uni-leipzig.de (I.G.); 2Institute for Research against Digestive Cancer (IRCAD), 67091 Strasbourg, France; michele.diana@ircad.fr; 3Department of General Surgery, Hospital Card. G. Panico, 73039 Tricase, Italy; 4Innovation Center Computer-Assisted Surgery (ICCAS), University of Leipzig, 04103 Leipzig, Germany; claire.chalopin@medizin.uni-leipzig.de (C.C.); hannes.koehler@medizin.uni-leipzig.de (H.K.); marianne.maktabi@medizin.uni-leipzig.de (M.M.); 5Institute of Pathology, University Hospital Leipzig, 04103 Leipzig, Germany; Katrin.Schierle@medizin.uni-leipzig.de

**Keywords:** hyperspectral imaging (HSI), colorectal cancer (CRC), machine learning, deep learning, optical biopsy, optical imaging

## Abstract

**Simple Summary:**

Detection of colorectal carcinoma is performed visually by investigators and is confirmed pathologically. With hyperspectral imaging, an expanded spectral range of optical information is now available for analysis. The acquired recordings were analyzed with a neural network, and it was possible to differentiate tumor from healthy mucosa in colorectal carcinoma by automatic classification with high reliability. Classification and visualization were performed based on a four-layer perceptron neural network. Based on a neural network, the classification of CA or AD resulted in a sensitivity of 86% and a specificity of 95%, by means of leave-one-patient-out cross-validation. Additionally, significant differences in terms of perfusion parameters (e.g., oxygen saturation) related to tumor staging and neoadjuvant therapy were observed. This is a step towards optical biopsy.

**Abstract:**

Currently, colorectal cancer (CRC) is mainly identified via a visual assessment during colonoscopy, increasingly used artificial intelligence algorithms, or surgery. Subsequently, CRC is confirmed through a histopathological examination by a pathologist. Hyperspectral imaging (HSI), a non-invasive optical imaging technology, has shown promising results in the medical field. In the current study, we combined HSI with several artificial intelligence algorithms to discriminate CRC. Between July 2019 and May 2020, 54 consecutive patients undergoing colorectal resections for CRC were included. The tumor was imaged from the mucosal side with a hyperspectral camera. The image annotations were classified into three groups (cancer, CA; adenomatous margin around the central tumor, AD; and healthy mucosa, HM). Classification and visualization were performed based on a four-layer perceptron neural network. Based on a neural network, the classification of CA or AD resulted in a sensitivity of 86% and a specificity of 95%, by means of leave-one-patient-out cross-validation. Additionally, significant differences in terms of perfusion parameters (e.g., oxygen saturation) related to tumor staging and neoadjuvant therapy were observed. Hyperspectral imaging combined with automatic classification can be used to differentiate between CRC and healthy mucosa. Additionally, the biological changes induced by chemotherapy to the tissue are detectable with HSI.

## 1. Introduction

Colorectal cancer (CRC) is the third most common carcinoma worldwide [1]. Despite rapidly developing treatment methods, early detection plays a key role in decreasing mortality [2]. Additionally, it is well-known that adenomas entail a 50% malignant transformation potential, and approximately one-quarter of them are missed during standard colonoscopy [3,4]. For this reason, a high-quality colonoscopy is fundamental in order to detect CRC and its potential precursors. Currently, histopathological analysis plays a critical role in assessing the malignancy potential of a lesion. However, this is time-consuming and expensive. For this reason, a number of new imaging techniques are implemented into flexible endoscopes in order to allow a so-called optical biopsy [5]. Such technologies are far from being a routine tool in clinical practice. Consequently, there is still a wide window of opportunity for research within this field.

Indeed, a tool capable of providing an instantaneous and reliable optical biopsy would have a great impact in the clinical practice, allowing to rapidly differentiate potentially premalignant lesions from CRC or benign lesions, thereby reducing the costs by avoiding sampling by means of multiple histological biopsies.

This novel tool might potentially have clinical relevance. First, it could be useful during the preoperative diagnostic stage to accurately detect tumor-bearing regions, and secondly, after cancer removal, in order to ensure tumor-free resection margins, which represent the milestone to achieve curative oncological results.

Hyperspectral imaging (HSI) is a contactless, contrast-free, and non-invasive optical imaging technology providing pixel-by-pixel spectroscopic and spatial information about the analyzed area. The tissue–light interaction generates specific spectral signatures, allowing tissue perfusion assessment and tissue differentiation [6,7,8,9,10,11,12,13]. HSI cameras are highly versatile and easily compatible with existing medical instruments, such as flexible endoscopes, otoscopes, and laparoscopes [14,15,16]. In the field of oncology, HSI has been successfully used to detect thyroid and salivary glands [17], gastric cancer [18], oral cancer [19,20], breast cancer [21,22], brain cancer [23,24], head and neck cancer [25,26,27], tumors of the kidney [28], as well as colon cancer [29,30,31,32,33,34] in humans.

To analyze and interpret the spectral data, traditional learning-based approaches such as support vector machines (SVMs), random forest (RF), and logistic regression (LR) as well as deep learning networks can be used. SVMs have been used to classify gastric cancer resectates [35] and colon cancer [29,30,31]; RF to classify oral cancer in vivo [36]; and LR, k-nearest neighbors (KNN), and neural networks for head and neck tumor classification [26]. The aim of the present study was to evaluate the potential of HSI to discriminate between healthy colonic mucosa, adenomas, and CRC using several machine learning (ML) approaches and statistical analysis methods.

## 2. Materials and Methods

### 2.1. Patient Cohort

This prospective, single-center, non-randomized, open-label, and single-arm clinical trial was performed at the University Hospital of Leipzig, Leipzig, Germany. The study was approved by the local ethical committee of the Medical School of the University of Leipzig (026/18-ek, 22 February 2018) and was registered at Clinicaltrials.gov (accessed on 22 February 2020) (NCT04230603). Written informed consent was obtained from all patients involved. All consecutive patients undergoing primary colorectal resections for CRC or endoscopically non-resectable adenomas were included from July 2019 to May 2020. Patients with cancer recurrence were excluded. Out of 67 patients, 7 patients had to be excluded because of lack of consent, and another 4 patients could not be examined due to HSI camera unavailability. Immediately after resection and still in the operating room, the specimen was divided ex-situ along the taenia libera (according to our pathologist’s internal guidelines), and the mucosal side was exposed. The mucosal side of the tumor-bearing bowel segment was imaged with the hyperspectral camera in a standardized fashion according to our standard operational procedures (SOPs; see below).

Fifty-four consecutive patients (*n* = 39 men; *n* = 16 women) with a median age of 66 years (range: 40–82) were included in this study. The operations were mainly performed laparoscopically, and 5 (6.3%) were open surgeries, due to considerable adhesions caused by prior operations.

### 2.2. Image Recording

HSI data were acquired with the TIVITA^®^ Tissue system (Diaspective Vision GmbH, Am Salzhaff, Germany), which has a spectral range of 500 to 1000 nm, a spatial range of 640 by 480 pixels, and an acquisition time of approximately 6 s. Measurements were performed under standardized conditions with ambient light turned off and a distance of 50 cm between the object and the HSI camera. By combining two-dimensional spatial data with a third spectral dimension, the system generated three-dimensional data called a hypercube. Under illumination with light in the visible and near-infrared spectrum, the analysis software provided one red green blue (RGB) and four false-color images with an effective number of 640 by 480 pixels. Measurements were performed at a 50 cm distance from the object, and an 8.0 by 6.5 cm^2^ field of view with a theoretical spatial resolution of 0.13 mm/pixel was achieved. A spatial resolution of 0.39 mm/pixel was set at 630 nm [37]. Additionally, the system generated false-color images, representing physiological tissue parameters, such as the tissue oxygenation (StO_2_) and the near-infrared perfusion index (NIR PI), the tissue water index (TWI), and the organ hemoglobin index (OHI). The parameters NIR PI, TWI, and OHI are specified in arbitrary units in a range from 0 to 100. The mentioned variables and their methods of determination have already been described in detail by Holmer et al. [38].

### 2.3. Image Annotation

An experienced pathologist (K.S.) together with an experienced surgeon (B.J.W.) annotated the RGB images manually. In cases of controversy, the images were compared to the histopathological slides, and a consensus was reached. Three main classified groups or classes were annotated in the pictures. The first class was cancer (CA), and the second class was the (potentially) adenomatous margin around the central tumor (AD). The third class was healthy mucosa (HM). These three areas were color-coded within the images. The remaining unannotated tissues did not undergo further processing (Figure 1).

### 2.4. Preprocessing and Classification

Based on the annotations, several structures were visualized, classified, and compared with one another:HM vs. CA;HM vs. CA with AD (CAAD);CA vs. AD.

Additionally, patient-specific factors such as the (yp)TNM classification of the resectate and the status/post-neoadjuvant therapy were analyzed.

To achieve high-quality outcomes, the spectral data were preprocessed. First, a Savitzky–Golay filter was used to smoothen the data [39]. The data were then normalized using a standard normalization transformation [40]. However, our dataset was imbalanced since a total of 2,795,571 spectra for HM and 340,973 spectra for CA merged with AD were annotated. In order to achieve high-performance results, a balancing of the data was achieved by randomly downsampling the most represented class (the HM class).

Several supervised classification frameworks (e.g., RF, SVM) were tested using Scikit-learn and Python (Python Software Foundation, version 3.7, www.python.org (accessed on 22 February 2020)) [41]. The best results were obtained with a neural network. A multi-layer perceptron (MLP) with a total of four layers and a hyperbolic tangent as activation function was implemented as a neural network. The MLP is a simple artificial convincing network that provides high performance in the classification of spectral data [42]. An improvement in the network was achieved by using a Gaussian filter on the HSI data to also use spatial information for classification purposes. To simulate a clinical environment, leave-one-patient-out cross-validation was performed to classify HM, CA, and AD. Finally, a visualization of the classified tissue structures was performed. In addition, a classification of patient-specific characteristics (e.g., previous neoadjuvant therapy, tumor stage) was performed by using stratified cross-validation with 2 folds. To measure the performance of the neural network, we calculated the following:$$Sensitivity=\frac{TP}{TP+FN}$$
$$Specificity=\frac{TN}{TP+FP}$$
$$Accuracy=\frac{TP+FP}{TP+FP+TN+FN}$$
where TP is the true positive (i.e., cancerous tissue correctly identified as cancerous tissue), FN is the false negative (i.e., cancerous tissue incorrectly identified as healthy mucosa), TN is the true negative (i.e., healthy mucosa correctly identified as healthy), and FP is the false positive (i.e., healthy mucosa incorrectly identified as cancerous tissue). The receiver operating characteristic (ROC) curve and the area under the curve (AUC) score were calculated for cancerous tissue.

### 2.5. Statistical Analysis of Physiological Parameters

The perfusion parameters of several tissue structures (e.g., HM) were analyzed by using statistical tests to show differences in the structures. Parameter images obtained from measurements and patient data information ((yp)TNM classification and status/post-neoadjuvant therapy) were inserted into Python (Python Software Foundation. version 3.7) for statistical analysis.

To calculate statistical differences between two datasets, we first used the Shapiro–Wilk test to evaluate a normally distributed population. Secondly, the sets were divided into independent and dependent samples. Afterward, if the population of the dataset was normally distributed, a Student’s t-test was performed for dependent and independent datasets. Otherwise, a Mann–Whitney and Wilcoxon signed-rank test was performed for independent and dependent samples, respectively.

## 3. Results

### 3.1. Patients

A total of 59 HSI records of 54 patients were obtained. Five additional incidental adenomas were found in five of the 54 procedures. The preoperative findings from these patients are shown in Table 1. Forty-nine (93.8%) operations were performed laparoscopically and five (6.3%) during open surgeries. The procedures performed included 32 anterior rectal resections (in 28 cases using the total mesorectal resection (TME) technique), 12 right- and 9 left-sided colon resections, and 1 transverse resection. The pathological evaluation revealed 48 carcinomas and 16 adenomas. Eighteen patients presenting with rectal cancer received neoadjuvant chemoradiotherapy. Two of the patients treated with neoadjuvant therapy had a complete tumor remission. Consequently, the CA class could not be annotated in these patients.

### 3.2. Classification and Visualization

Three leave-one-patient-out cross-validations were performed: (i) CAAD vs. HM, (ii) CA vs. HM, and (iii) CA vs. AD.

The classification result for the study with CA vs. HM without AD achieved better results (AUC = 97, sensitivity = 86%, specificity = 95%) than CAAD vs. HM (AUC = 95, sensitivity = 89%, specificity = 88%) (Figure 2). However, CA against AD reached an AUC score of 71, a sensitivity of 68%, and a specificity of 59% for cancerous tissue (Figure 2). The classification and visualization of malignant tissue and healthy mucosa were performed in less than 5 s (Figure 3).

In summary, in the CA vs. HM without AD analysis, a sensitivity of less than 50% was achieved only for 13% of patients. The specificity was more than 68% for all patients. The sensitivity was low for patients with AD and CA in (y)pT1 and (y)pT2 tumor stages, and the specificity was low for ypT3 and ypT4 tumor stages as well as in patients previously treated with neoadjuvant therapy.

Additionally, a differentiation between CA, which had been treated with neoadjuvant therapy (N = 18), and CA, which had not been treated with neoadjuvant therapy (N = 11), was evaluated. An AUC score of 97 and a sensitivity and specificity of 92% for detecting CA, which was treated by neoadjuvant therapy, were achieved. 

A comparison between the tumor stage of group 1, which included (y)pT1 and (y)pT2 (*n* = 19), and the tumor stage of group 2, which included (y)pT3 and (y)pT4 (*n* = 27), was made. Regarding the determination of tumor stage, group 2 ((y)pT-stage, (y)pT3, and (y)pT4) achieved a sensitivity of 92% and a specificity of 90%, as well as an AUC score of 97.

### 3.3. Data Analysis of Physiological Parameters

Statistical tests were performed to analyze the difference between perfusion and water content indices of the tissue structures (Table 2). These tests showed that after separation of the tissue structures into with and without status/post-neoadjuvant therapy, the divergence of the perfusion parameter of these several tissue structures, e.g., the TWI, OHI, and NIR PI between (y)pT1-(y)pT2 CA vs. HM, changed (Table 2).

HM showed lower TWI, OHI, and StO_2_ than CA (Figure 4A,B, Table 2).

Additionally, the comparison of tumor stages (y)pT1 and (y)pT2 together against (y)pT3 and (y)pT4 together showed significant differences for water content (in patients receiving neoadjuvant therapy) and NIR-PI (see Figure 4C, Table 2).

AD showed a higher TWI, OHI, and StO_2_ than HM, whereas CA had a higher TWI and StO_2_ than AD.

## 4. Discussion

The results of our study clearly demonstrate that hyperspectral imaging combined with artificial intelligence techniques make it possible to sharply differentiate CRC from adenoma and healthy mucosa (AUC score of 97). This represents a step forward in hyperspectral-based automatic tissue recognition. Such a technological advance could have a great clinical impact and has the potential to provide the endoscopist with a powerful new tool. In particular, in the near future, it will make it possible to diagnose a tumor or tumor infiltration within the resection margin after endoscopic removal of superficial (mucosal or submucosal) cancer or adenoma in real-time. Additionally, in our analyses, HSI revealed an increased perfusion within cancerous tissue as compared to healthy mucosa. This finding might be explained by the new vessel formation often encountered in CRC as a consequence of the hyperproduction of vascular endothelial growth factor (VGEF) [43]. On the other hand, advanced cancers ((y)pT3-T4) showed a significantly lower perfusion than less advanced ones ((y)pT1-T2). This observation might result from the large central necrotic areas frequently present within large tumoral lesions. Our observations confirm the great potential of HSI to disclose tissue physiology, as observed in previous experimental studies [6,44]

Previously, the groups of Baltussen et al. and Beaulieu et al. were able to distinguish cancer from healthy tissue successfully [29,30]. Baltussen et al. only achieved an AUC score of 81, using an HSI system working in a wider spectral range than our study (ranging from 400 to 1000 nm) [29]. The authors used two different HSI cameras simultaneously to analyze the specimens of 54 patients (400–1000 and 900–1700 nm). Interestingly, when using the camera with a similar spectral range to that used in our work, they noticed a consistent drop in the accuracy of the ML algorithm. The reduction of performance noted by the authors when using one HSI camera could be explained by the fact that they acquired HSI data a long time after specimen removal from the abdomen. This might have resulted in extensive cellular death (apoptosis) within the healthy and cancerous mucosa, impairing the precise discrimination of the two types of tissues. Although our dataset was acquired immediately after extraction from the human body and without any specimen processing, this corresponds to very similar in vivo conditions. Despite the remarkable results of Baltussen et al., the simultaneous use of two HSI systems results in a long acquisition time (approximately 50 s) when compared to our study (about 6 s), which can be suitable for experimental purposes but certainly represents a burden for daily clinical practice [29].

Beaulieu et al. achieved a sensitivity of approximately 92% and a specificity of 90% by using 15 patients. In comparison to our study, the sensitivity was 6% higher, despite 5% less specificity using *n* = 42 fewer patients than in our study and a broader wavelength range from 350 to 1800 nm [30]. However, the authors used probe-based spectroscopy, which analyzed only pinpoint areas, not allowing for spatial localization. Precise intraoperative spatial localization is fundamental to identifying cancerous tissue prior to its removal. As a result, probe-based spectroscopy largely impairs intraoperative usability. Importantly, we evaluated the performance of the algorithm by using a-leave-one-patient-out cross-validation. Beaulieu et al. instead used 50% of the data for training and 50% for testing, and this type of approach led to an overestimation of the classification’s accuracy [30].

In our current study, the specificity and sensitivity of each patient were considered, which clearly showed that adenoma, different tumor stages, as well as the previous treatment with neoadjuvant therapy influence classification outcomes. The differences between tissue properties of AD and HM are lower than between CA and HM, possibly due to the lower sensitivity of the adenoma in the study of classification CAAD vs. HM. We assume that ypT3 and ypT4 tumor stages affected the HM more than (y)pT1 and (y)pT2 tumor stages, due to the lower specificity for ypT3 and ypT4 tumors in the classification study using a neural network.

The low specificity of patients who had neoadjuvant therapy can be explained by the biological modifications altering the tissue’s intrinsic features following chemotherapy. Due to the high differences within perfusion parameters as well as the high performance in detecting tissue after neoadjuvant therapy (AUC = 97), we assume that in the future HSI could help to quantify the tumor’s response to neoadjuvant therapy. One of the advantages of the HSI system that we used was that it measured several physiological parameter indices as an immediate output. For the first time, tissue parameters of cancerous colorectal resectates were measured intraoperatively using a spectral imaging system. Significant effects of neoadjuvant therapy were shown, in the case of comparison between different tumor stages and tissue types. For example, TWI and the NIR-PI are lower for ypT3-T4 cancer receiving neoadjuvant therapy than pT3-pT4 cancer with and without receiving neoadjuvant therapy (Table 2). An accurate assessment of the tumor response using non-invasive perfusion imaging of cell death will offer an even greater benefit. To date, in assessing tumor perfusion and treatment response, computed tomography (CT), magnetic resonance imaging (MRI), and hybrid positron emission tomography (PET) have been used [45]. These techniques obtain perfusion, oxygenation, and glucose consumption. In future studies, the applicability of HSI to analysis of tumor characterization, as well as the patient’s individual response to anticancer drugs, can be evaluated.

In our study, the physiological parameters of several tumor stages were considered. It should be noted that we examined resected probes. However, we can assume that this factor did not have a strong impact on perfusion parameters, as can be seen in Figure 5. This is possibly due to the immediate data acquisition after extraction from the body and without any specimen processing, very much resembling the in vivo condition (only very few seconds).

Near-infrared (NIR) fluorescence imaging with indocyanine green (ICG) can also be used to show perfusion parameters. In colorectal surgery, this imaging modality is used to reduce the risk of anastomotic complications [46]. Detection of cancer using NIR fluorescence intraoperative imaging has also been reported to be useful in the literature [47,48,49]. Unlike HSI, which is a contrast-free technique, NIR fluorescence imaging requires the intravenous administration of a contrast agent (ICG). Side effects related to intravenous ICG, such as anaphylactic shock, drop in blood pressure, and tachycardia have been pointed out. Additionally, in the majority of the studies using near-infrared fluorescence imaging, perfusion is measured subjectively and is not quantified.

An important benefit of HSI technology is that it is very easy to use in operating rooms, and commercially available HSI systems with medical certification already exist. However, HSI limitations concerning penetration into biological tissue exist (e.g., at 850 nm: 3.75 mm) [25]. A further drawback of HSI is the lack of a current laparoscopic system, although a rapid technological development brought about by the new sensor technology has been described recently [7]. Additionally, HSI has already been integrated in a laparoscopic system and in a flexible endoscope to achieve optical tissue examination probes [14,16,50]. The disadvantages of these technologies are that real-time acquisition of high resolution HSI cube analysis has not been implemented yet. With HSI technologies, temporal, spectral, and spatial resolution [7] must be considered. It is essential to mention here that a large quantity of data from different patients is essential for excellent results when using artificial intelligence methods with spectral data from biological tissue. If the spectral range is moreover restricted, larger quantities of data are required. The extension of the analysis in the wavelength range showed increased performance of automatic classification methods [29]. New insights into the tissue can be expected due to deeper penetration depth [25]. If real-time acquisition is not required in clinical practice, larger wavelength ranges and high spectral resolutions, like in our study (spectral range: 500 to 1000 nm, resolution: 5 nm), can be used. Such endoscopic HSI technology was described in Köhler et al. [14]. Fast acquisition time for the HSI data (within 4.6 s) was obtained so that our trained model could be embedded with the system. As a result, in future applications of spectral endoscopic techniques, the findings of our study could be used to implement innovative spectral technologies that automatically differentiate and analyze tumors.

## 5. Conclusions

Hyperspectral imaging is a promising new tool that allows cancer recognition. However, research in this field is still at a preliminary stage. In this study, a realistic surgery setting was applied. A high accuracy of 94% was achieved in the classification of cancerous tissue at the mucosal side combining HSI with a neural network. The future development of HSI-based systems performing contactless and non-invasive optical biopsies of in vivo tissue will benefit from the results of this work.

## Figures and Tables

**Figure 1 cancers-13-00967-f001:**
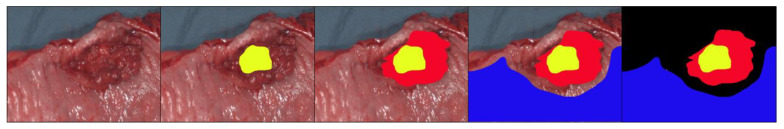
Annotation process. A pT3 adenocarcinoma of the sigmoid colon. The first annotation in yellow marks the certain tumor tissue; in red the surrounding tissue, probably with adenoma parts; in blue the healthy mucosa; and in black the deleted unmarked areas.

**Figure 2 cancers-13-00967-f002:**
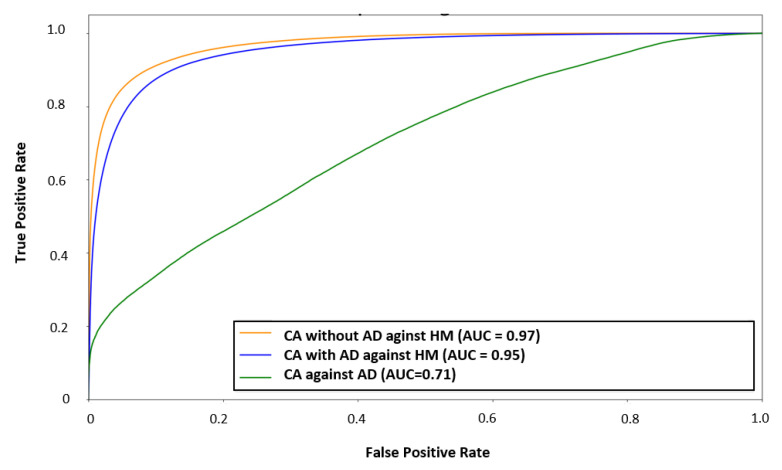
Averaged receiver operating characteristic (ROC) curve for every patient’s individual evaluation for classification: Cancer (CA) with adenomatous margin around the central tumor (AD) against healthy mucosa (HM), CA without AD against HM, and CA against AD (AUC—area under the curve).

**Figure 3 cancers-13-00967-f003:**
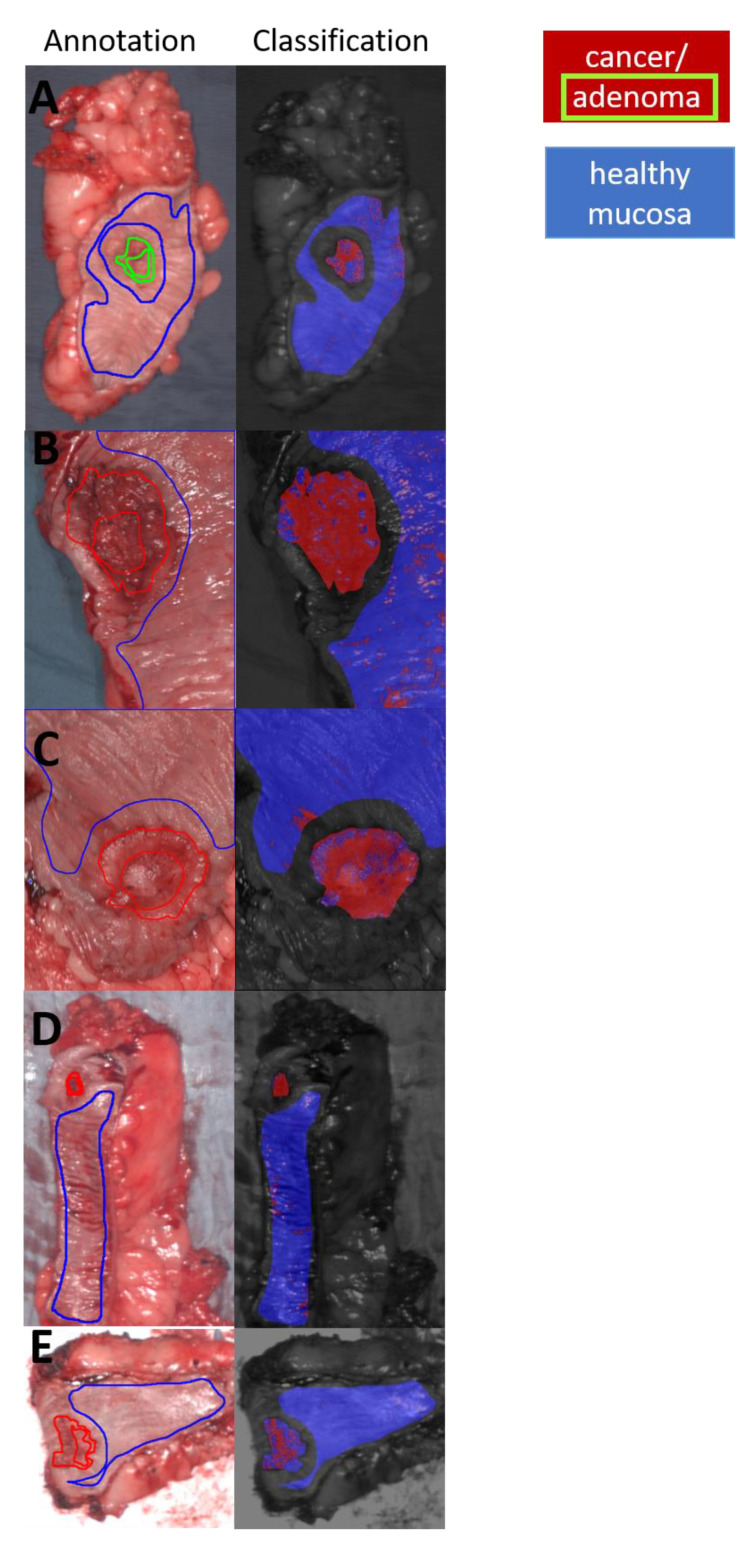
(**A**,**C**,**E**,**G**) represent annotation and classification of cancerous tissue: A = adenoma; B = ypT2 pN1a (1/23) pM1a (PUL) L1 V0 Pn0, UICC Stage IV A; C = pT2 pN0 (0/18) M0 L0 V0 Pn0, UICC Stage I; D = pT3a pN0 (0/29) M0 L1 V0 Pn0, UICC Stage IIA; E = ypT3 ypN1b (2/21) M0 L1 V0 Pn0, UICC Stage yp III A. (red fill: cancer and adenoma, red line: cancer, green line: adenoma, blue fill: healthy mucosa, blue line: healthy mucosa).

**Figure 4 cancers-13-00967-f004:**
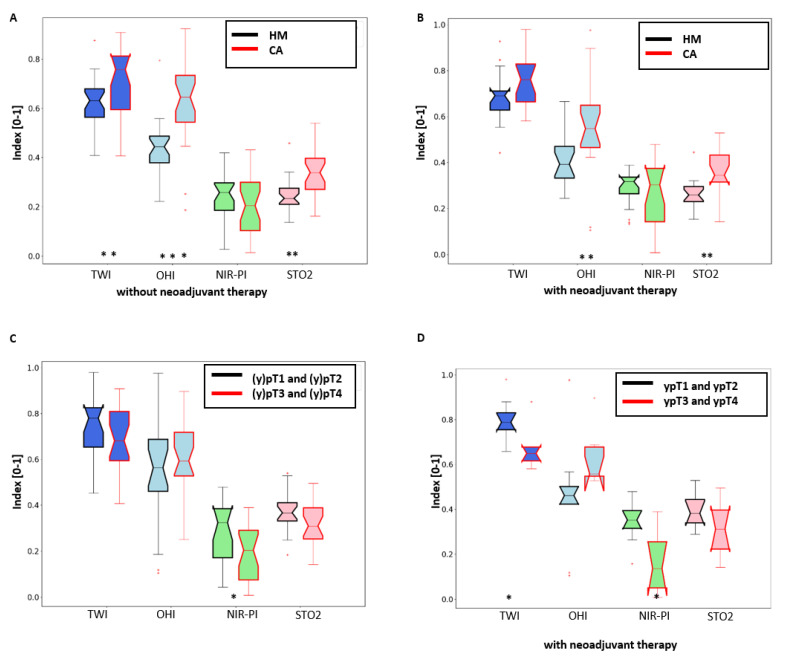
Comparison between physiological parameters TWI, OHI, NIR-PI, and StO_2_. (**A**) Comparison of healthy and cancerous tissue without neoadjuvant therapy. (**B**) Comparison of healthy and cancerous tissue with neoadjuvant therapy. (**C**) Comparison of cancerous tissue: (y)pT1 and (y)pT2 stage vs. (y)pT3 and (y)pT4 stage. (**D**) Comparison of cancerous tissue: ypT1 and ypT2 stage vs. ypT3 and ypT4 stage with neoadjuvant therapy.

**Figure 5 cancers-13-00967-f005:**
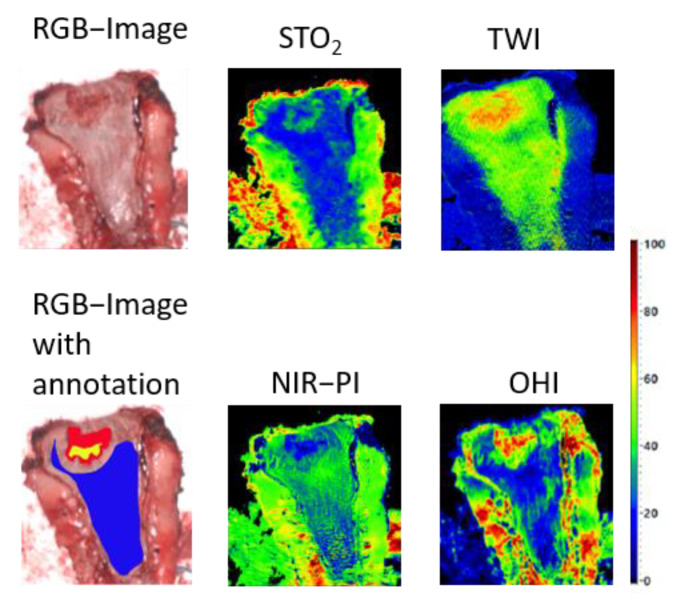
An example of tissue specimen with a tumor (ypT2). Tissue structures show homogeneous physiological parameter values (blue: healthy mucosa, yellow: cancer, red: tumor margin).

**Table 1 cancers-13-00967-t001:** Patient and tumor characteristics.

Variables	Number
Patients	54
Male/female	37/17
Pathologies	59
Colon/rectum	22/37
Adenoma (colon/rectum)	11 (6/5)
Unsuspected adenoma	5
Carcinoma (colon/rectum)	48 (16/32)
Neoadjuvant treatment	18
ypT0	2
pT1, ypT1	3
pT2, ypT2	16
pT3, ypT3	25
pT4, ypT4	2

**Table 2 cancers-13-00967-t002:** Statistical comparison of the mean physiological parameter indices computed for different kinds of tissues (HM, AD, CA) and regarding different characteristics (neoadjuvant therapy, tumor classification); blue = no statistical difference *p* > 0.05; red = * *p* ≤ 0.05; yellow = ** *p* ≤ 0.01; white = *** *p* ≤ 0.001. The symbol # denotes independent datasets.

Tissue	Tissue-/Tumor Classification	TWI	OHI	NIR-PI	StO_2_
CA	#CA: (y)pT1-(y)pT2 vs. (y)pT3-(y)pT4	(0.78/0.68)	(0.59/0.56)	(0.32/0.2) *	(0.37/0.31)^0.061^
-	#CA: ypT1-ypT2 vs. ypT3-ypT4	(0.79/0.65) *	(0.46/0.56)	(0.35/0.14) *	(0.38/0.31)
CA vs. HM	CA ((y)pT) vs. HM	(0.76/0.65) *	(0.59/0.42) ***	(0.22/0.26)	(0.34/0.24) ***
-	CA (pT) vs. HM	(0.76/0.63) **	(0.65/0.44) ***	(0.2/0.26)	(0.34/0.23) **
-	CA (ypT) vs. HM	(0.76/0.69)^0.082^	(0.55/0.39) **	(0.3/0.32)	(0.34/0.26) **
-	pT1-pT2 CA vs. HM	(0.67/0.62)	(0.68/0.46) **	(0.19/0.27)	(0.35/0.26)^0.055^
-	ypT1-ypT2 CA vs. HM	(0.79/0.69) *	0.46/0.33)	(0.35/0.32)^0.093^	(0.38/0.24) ***
-	pT3-pT4 CA vs. HM	(0.76/0.64) *	(0.6/0.43) **	(0.21/0.24)	(0.31/0.23) *
-	ypT3-ypT4 CA vs. HM	(0.65/0.71)	(0.56/0.42) *	(0.14/0.31) **	(0.31/0.27)
AD	#AD vs. HM (from (y)pT1-(y)pT2)	(0.71/0.66)	(0.55/0.4) *	(0.32/0.31)	(0.33/0.26)
-	#AD vs. HM (from (y)pT3-(y)pT4)	(0.71/0.65)	(0.55/0.43) *	(0.32/0.27)	(0.33/0.24) *
-	#AD vs. HM (from ypT)	(0.71/0.69)	(0.55/0.39) *	(0.32/0.32)	(0.33/0.26)
-	AD vs. HM	(0.71/0.65)	(0.55/0.42)^0.052^	(0.32/0.27)	(0.33/0.24) *
-	#AD vs. ypT1-ypT2 CA	(0.71/0.79)	(0.55/0.46)	(0.32/0.35)	(0.33/0.38)
-	#AD vs. HM (from ypT1-ypT2)	(0.71/0.69)	(0.55/0.33) *	(0.32/0.32)	(0.33/0.24) *
-	#AD vs. HM (from ypT3-ypT4)	(0.71/0.71)	(0.55/0.42) *	(0.32/0.31)	(0.33/0.27)

## Data Availability

Not applicable.

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
