# Peer review of "Feedforward Artificial Neural Network-Based Colorectal Cancer Detection Using Hyperspectral Imaging: A Step towards Automatic Optical Biopsy"

_cancers, 2021, doi:10.3390/cancers13050967_

Round 1
Reviewer 1 Report
A very fine and well written manuscript. Please clarify the is a Ex Vivo study in the text and discuss the disadvantages and advantages of implementation of the results in live examinations of patients.
Please clarify and add more details in the section describing the statistical analysis, which method is usef for which material.
Reviewer 2 Report
The main treatment for localized primary and early stage colon cancer is complete resection of the tumor. The success of the surgical intervention relies on accurate detection of malignant tumor boundaries. Visual inspection may lead to interpretation errors. Hyperspectral imaging (HSI) is a non-ionizing sensing technique with the ability to capture the diffuse reflectance spectra across the visible and near-infrared wavelength range. The potential of this rising technology is mainly to provide a label-free intra-operative feedback to the surgeon for objective assessment of cancer. Several data are today available among HSI potentiality and applications in cancer diagnosis.
In this paper the Authors investigated the capability of HSI in identifying tumour tissues from healthy mucosae and adenomas, concluding that it could be a promising new tool which allows cancer recognition.
In my opinion this topic is interesting and reinforces the data already known. The experimental design is well conducted, the results of the experiments performed support the conclusions and the paper is quite clearly written.
To improve their work, I suggest the Authors discuss the possible limits of this technology. In my opinion, this could help to translate their research in clinical practice.
I encourage the authors to continue their studies, in light with what they have suggested, by carrying out a larger study that could clearly prove the HSI applicability in CRC diagnosis during cancer surgery.
I recommend the Authors to carefully read the paper to revise language misspellings.
Reviewer 3 Report
This is a very interesting study that aims to evaluate the potential of HSI to discriminate between healthy colonic mucosa, adenomas, and CRC, using several machine learning (ML) approaches and statistical analysis methods.
General comments:
- I commend the effort by the authors in trying to evaluate a new technology for better characterizing this difficulty and challenging question.
- I think the study is very well done and includes all the details require for a good manuscript.
